# Complement-mediated killing of *Escherichia coli* by mechanical destabilization of the cell envelope

Georgina Benn [ID] [1,2,7], Christian Bortolini [ID] [1,3,7], David M Roberts [ID] [4], Alice L B Pyne [ID] [1,5], Séamus Holden [ID] [4] & Bart W Hoogenboom [ID] [1,6] ✉

## Abstract

Complement proteins eliminate Gram-negative bacteria in the blood via the formation of membrane attack complex (MAC) pores in the outer membrane. However, it remains unclear how outer membrane poration leads to inner membrane permeation and cell lysis. Using atomic force microscopy (AFM) on living *Escherichia coli* (*E. coli*), we probed MAC-induced changes in the cell envelope and correlated these with subsequent cell death. Initially, bacteria survived despite the formation of hundreds of MACs that were randomly distributed over the cell surface. This was followed by larger-scale disruption of the outer membrane, including propagating defects and fractures, and by an overall swelling and stiffening of the bacterial surface, which precede inner membrane permeation. We conclude that bacterial cell lysis is only an indirect effect of MAC formation; outer membrane poration leads to mechanical destabilization of the cell envelope, reducing its ability to contain the turgor pressure, leading to inner membrane permeation and cell death.

**Keywords** Complement; Membrane Attack Complex; *Escherichia coli*; Atomic Force Microscopy; Bacterial Membranes
**Subject Categories** Immunology; Microbiology, Virology & Host Pathogen Interaction

## Introduction

Gram-negative bacteria are protected by a cell envelope that consists of an outer and inner membrane, separated by the peptidoglycan cell wall (Silhavy et al, 2010; Lithgow et al, 2023). This multilayer protection is an important factor in bacterial resistance against antibiotics (Delcour, 2009). Its multilayer nature also implies complex mechanical behavior: whereas it has long been established that the peptidoglycan cell wall is a key player in cell envelope mechanics, it has only recently been shown that the outer

membrane also contributes to defining how bacteria support mechanical stress (Rojas et al, 2018; Sun et al, 2022; Fivenson et al, 2023).

To breach the cell envelope and eliminate Gram-negative bacteria in the bloodstream, the complement system contains five proteins, C5–C9, that directly participate in bacterial killing (Muller-Eberhard, 1986; Ricklin et al, 2010; Doorduijn et al, 2019). Activation of complement leads to cleavage of C5—into C5a and C5b—by C5 convertases at the bacterial surface; C5b next oligomerizes with C6, C7, C8 and multiple copies of C9 to form membrane attack complex (MAC) assemblies (Fig. 1A). The C5b–C8 complex is critical for MAC function; in its absence, C9 may oligomerize yet does not bind to or perforate membranes (Dudkina et al, 2016; Parsons et al, 2019). The C5b–C9 MAC pores are tall enough to traverse a single membrane (Serna et al, 2016; Menny et al, 2018; Sharp et al, 2016) but not the entire Gram-negative bacterial cell envelope. Moreover, it can be excluded that MACs are formed in the inner membrane, since bacteria can be lysed by C9 when the only available C5b–C8 is bound to the outer membrane (Heesterbeek et al, 2019). In other words, the direct contribution of MACs to lysis only occurs at the outer membrane.

Recent years have witnessed substantial progress in our structural understanding of isolated MACs (Dudkina et al, 2016; Serna et al, 2016; Menny et al, 2018) and of MACs in phospholipid model membranes (Parsons et al, 2019; Sharp et al, 2016). In addition, studies of complement have been facilitated by protocols that expose bacterial cells to the MAC under semi-purified conditions; under such conditions, MAC pores were also visualized at the bacterial surface by AFM; this detection of MACs on single cells correlated with outer membrane perforation and cell death in population assays (Heesterbeek et al, 2019; Doorduijn et al, 2020). Hence MAC formation and cell death can be uniquely attributed to C5–C9, without requiring other serum components, for example, lysozymes to degrade the peptidoglycan cell wall (Heesterbeek et al, 2019; Doorduijn et al, 2020).

Nonetheless, the fundamental question remains how MAC formation in the outer membrane leads to inner membrane permeation and thereby to cell lysis and death (justified by previous work (Heesterbeek et al, 2019), we will use the terms inner membrane permeation and cell lysis/death interchangeably). While

[1]London Centre for Nanotechnology, University College London, London WC1H 0AH, UK. [2]Department of Molecular Biology, Princeton University, Princeton, NJ 08544, USA. [3]National Physical Laboratory, Hampton Road, Teddington TW11 0LW, UK. [4]School of Life Sciences, University of Warwick, Gibbet Hill Campus, Coventry CV4 7AL, UK. [5]Department of Materials Science and Engineering, University of Sheffield, Sheffield S10 2TN, UK. [6]Department of Physics and Astronomy, University College London, London WC1E 6BT, UK. [7]These authors contributed equally: Georgina Benn, Christian Bortolini. ✉E-mail: b.hoogenboom@ucl.ac.uk

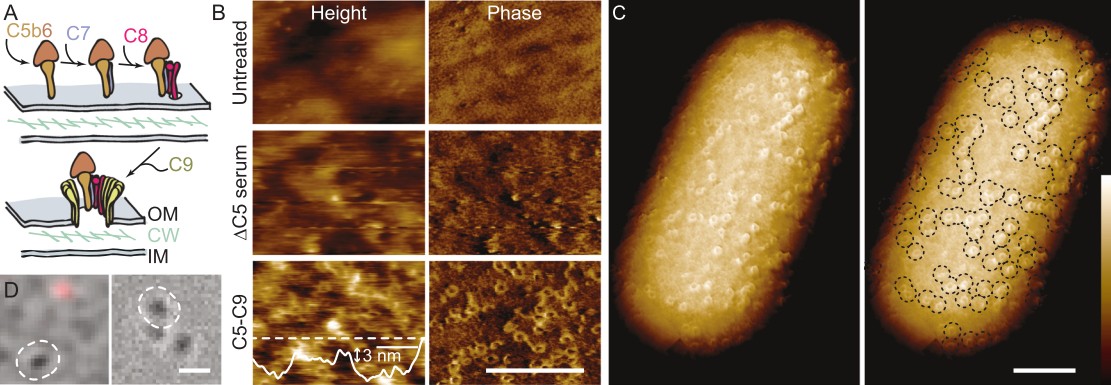

**Figure 1. Bacteria resist lysis by complement despite extensive MAC poration of the outer membrane.**

(A) Schematic of MAC formation, with formation of the C5b–C8 complex and next poration of the outer membrane by oligomerization of C9 (OM outer membrane, CW cell wall, IM inner membrane). (B) High-magnification AFM height and phase images on the surface of untreated, of ΔC5 serum treated, and of full MAC treated BL21(DE3) *E. coli*. The outer membrane before MAC treatment is smooth, with a background of outer membrane proteins in a hexagonal lattice faintly visible in the phase image (Benn et al, 2021). Addition of ΔC5 serum components activates the MAC pathways up to the C5 convertase and coats the outer membrane with complement proteins. Addition of C5-9 leads to MAC pores that protrude ~3 nm against a corrugated background, as measured by line profile along dashed white line. (C) 500 nm wide phase images were tiled to provide a view of the entire cell. For the 3D representation shown here, flattened images were tiled along the overall cell curvature. Right panel has MACs marked by dashed black lines. (D) Merge of brightfield (greyscale) and SYTOX™ fluorescence (red) images of bacteria. White dashed line in left panel shows the cell imaged in (B) (C5–C9) and right panel shows the cell imaged in (C). Color scale (see inset, C): 10 nm (B, height) and 1.5, 2.5 and 3.5 degree (B, phase), 1.5 degree (C). Scale bars: 200 nm (B, C), 3 μm (D).

multiple hypotheses have been proposed (Doorduijn et al, 2019), the different possible pathways remain hard to disentangle. Of note, in bulk assays, it is nigh impossible to reliably access the temporal window between outer membrane poration and inner membrane permeation, as complement action is unlikely to be synchronized across whole bacterial populations.

To establish the relationship between MAC poration of the outer membrane and permeation of the inner membrane, we therefore used single-cell AFM (Viljoen et al, 2020) to characterize MAC-induced changes in the structure and mechanics of the bacterial surface while monitoring the viability of these individual cells by the influx of small molecules into the cytoplasm, using SYTOX™ staining (Benn et al, 2019, 2021) (Fig. 1).

## Results

Following previously established protocols (Heesterbeek et al, 2019; Doorduijn et al, 2020), we primed BL21(DE3) and MG1655 *E. coli* for MAC assembly by exposure to C5-deficient (ΔC5) or C8-deficient (ΔC8) serum, followed by washing. This led to the deposition of C5 convertases and, for ΔC8 serum, to the assembly of C5b-C7, but leaving the outer membrane intact. To complete MAC assembly, the cells were incubated with purified complement proteins: C5–C9 and C8–C9 after exposure to ΔC5 and ΔC8 serum, respectively.

Imaging living cells using AFM AC mode with both topography ("height") and phase imaging, we detected circular assemblies on the bacterial surface, consistent in size and morphology with MAC pores as characterized previously (Parsons et al, 2019; Serna et al, 2016; Menny et al, 2018; Sharp et al, 2016; Heesterbeek et al, 2019) (Fig. 1B,C). The phase images provided contrast based on local material properties (García and Pérez, 2002) and facilitated the

detection of protein assemblies against more complex backgrounds (Benn et al, 2021). The AFM surface topography ("height") was consistent with the phase images, albeit with a contrast that was compromised by the presence of nanometer-scale corrugation of the outer membrane. The deposition of proteins by serum obscured the underlying network of outer membrane proteins, as previously observed on untreated bacteria (Fig. 1B and ref (Benn et al, 2021)).

MACs here appeared to protrude ~3 nm from the surface (Fig. 1B). This is smaller than the ~10 nm height of the MAC pore rim as observed above a reconstituted phospholipid membrane surface (Parsons et al, 2019; Sharp et al, 2016). However, on cells the MAC height is not measured with respect to the lipid head groups, but with respect to a lipopolysaccharide cell surface decorated with serum components, reducing the apparent protrusion of the MACs above the cell surface; and under the conditions used here, we cannot exclude some compression by the AFM tip either. Our AFM experiments also revealed substantial cell-to-cell variability in the number of MACs on the cell surface (Appendix Fig. S1), but no significant preference for their locations on the cell surface (Appendix Fig. S2). While some images suggested local clustering of MACs, statistical analysis revealed no clustering beyond what may be expected for random deposition (Appendix Fig. S3).

Surprisingly, we found that individual cells could show no staining for inner membrane permeation (Fig. 1D) despite their outer membrane being perforated by tens to hundreds of MAC pores per μm² (Appendix Fig. S1). This demonstrates that extensive outer membrane disruption alone is insufficient to perturb the integrity of the cytoplasm and thereby insufficient to directly cause cell death.

To determine how MAC poration causes cell death, we next monitored changes at the bacterial surface and inner membrane permeation as a function of time. For cells that were initially alive

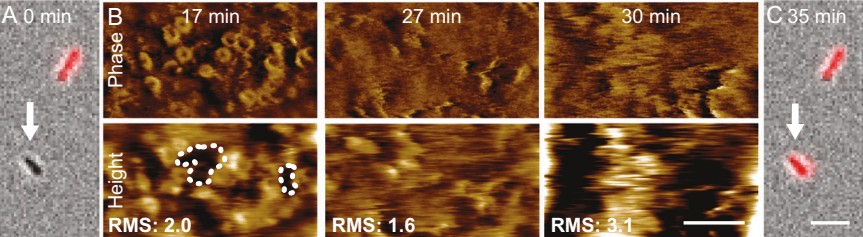

**Figure 2.  Inner membrane permeabilization correlates with destabilization of the bacterial surface.**

(A) Merge of brightfield (gray) and SYTOX™ (red) images of BL21(DE3) *E. coli* at the beginning of imaging (= 0 min). (B) AFM phase and height images of the cell that is marked by the white arrow in A, as a function of time, showing loss of contrast (quantified by an increase in RMS from 1.6 to 3.1 nm) at ~30 min, presumably due to disruption of the cell envelope. White dashed lines in the left hand image indicate what appear to be larger defects at the surface become harder to discern when the overall surface is disrupted in subsequent images. (C) As (A), the cell is SYTOX™ positive (red) when next inspected for inner membrane permeation. Color scale (see inset Fig. 1C): 2 deg and 10 nm (B). Scale bars: 5 µm (A, C) and 100 nm (B). Time points are relative to the initial image (A).

(Fig. 2A), MAC appeared again as circularly assemblies, readily identified in the phase images (Fig. 1) and by protrusions at according locations in the height images. However, the background in the height images also suggested gaps in the membrane—showing as black holes in the height images—that were substantially larger than the MACs, here with widths of the order of 100 nm (Fig. 2B, 17 min, height image). Furthermore, cells became harder to image with high resolution after cell death, but the general shape was still apparent (Fig. 2; Appendix Fig. S4). Viewed at higher magnification, this was apparent by a progressive blurring of features on the outer membrane, a drastic change in MAC positions and increase of the overall corrugation of the surface (Fig. 2B, 27 and 30 min). When the fluorescence signal for this cell was next monitored again (Fig. 2C, 38 min), the cell was found dead.

Continuously monitoring the corrugation and SYTOX™ staining, we found that this change in MAC positions and increase in corrugation directly preceded inner membrane permeation and thereby cell death (n = 3; Appendix Fig. S5). To probe if this was a generic feature of cell death, we carried out similar experiments on bacteria exposed to the antimicrobial peptide melittin, which targets bacterial membranes (Pan and Khadka, 2016). This did not result in such signatures of mechanical disruption upon cell death (Appendix Fig. S6), suggesting that the observed mechanical disruption is specific for MAC-induced killing.

Since AFM is a surface scanning technique, it requires a stable, static sample. Abrupt, diffuse changes in a surface, seen in Fig. 2 and Appendix Fig. S5, are indicative of detachment from an underlying support structure or readier mechanical dislodgement by the scanning AFM tip. This loss of contrast cannot be attributed to a general drop in AFM quality, since image quality is still good on other cells in the same field of view, but not SYTOX™ stained (Appendix Fig. S4). Noting that the general shape of cells can still be detected by AFM and brightfield microscopy (Appendix Fig. S4), these observations suggest that the cell envelope is becoming unstable and that this is part of the process leading to cell death.

Focussing on the pore formation and wider disruption of the outer membrane, we examined defects at cellular length scales, finding extended areas over which the outer membrane appeared to be removed (Fig. 3A, darker and therefore lower areas at the cell surface). For a clearer and higher-magnification view of these defects, we used AFM imaging based on fast force spectroscopy

(Dufrêne et al, 2017). Although slower and lacking the phase contrast of AC/dynamic mode imaging, this yielded stabler imaging conditions at the bacterial surface, even after cell death (Fig. 3B; Appendix Figs. S7–S9; Movies EV1 and 2). As before, MACs appeared as crater-shaped protrusions at the surface, appearing yellow to white with the color scale used here, but the outer membrane background was now better resolved.

In addition to these MAC protrusions, larger defects in the outer membrane background were clear (appear dark brown to black here), in some cases connected with MAC pores (Appendix Fig. S7) and seen to increase in number and in size as a function of time (dark areas in Fig. 3B). This progressive defect formation took place without the assembly of additional MAC pores: bacteria were washed after complement exposure and MAC formation, hence no proteins were available to generate additional MACs; indeed we did not observe an increase in the number of MAC assemblies over the image sequence (Fig. 3B). On the other hand, and surprisingly, this defect formation occurred on cells with intact inner membranes (Fig. 3C,D). That is, the outer membrane was disrupted by defects that extended over 100 s of nanometers, while the inner membrane appeared intact and the cell alive. By contrast, when these large-scale defects were observed on dead cells, their progression was halted (Appendix Fig. S9).

Taken together, these observations reveal the following sequence of events: (i) MAC pores assemble in the outer membrane; (ii) the structural and mechanical integrity of the outer membrane is progressively disrupted, at least in part via the formation of defects that extend far beyond and may be initiated by MAC pores; (iii) the inner membrane is permeated, and the cell dies.

To better understand the role of mechanics in this sequence of events, we recorded force-versus-indentation curves on complement-treated cells (see "Methods"), using AFM tips of a diameter that is large (130 nm) compared to the MACs and thereby more sensitive to overall changes in bacterial surface mechanics (Rojas et al, 2018), rather than to poration or other local defects. From these measurements, we determined the un-indented cell shape and, for indentations <200 nm, the surface mechanics over larger areas of the cell surface (Fig. 4A,B; Appendix Figs. S10–S12).

For dying cells, we observed an increase in cell size of 5–10% as a function of time, quantified via the height (diameter) of complement-treated BL21(DE3) and MG1655 *E. coli* (Fig. 4C,D). This swelling occurred after complement exposure and washing,

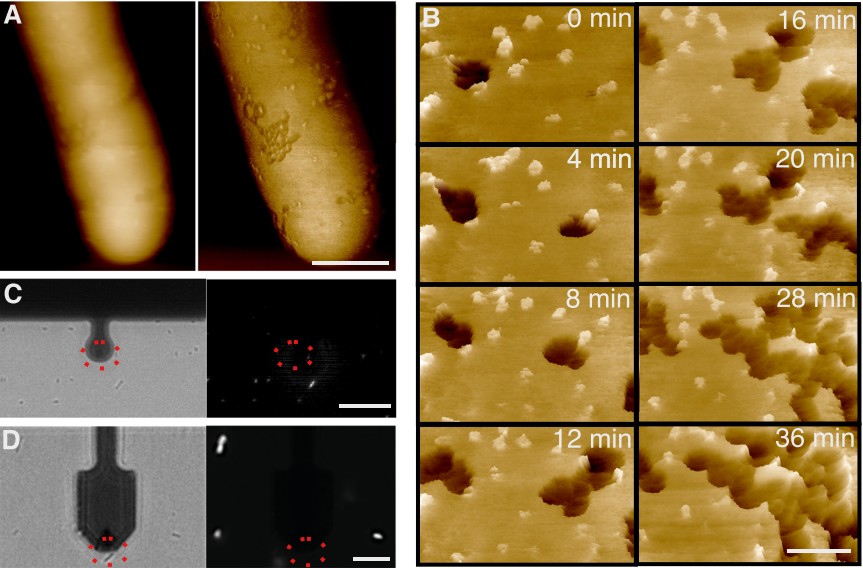

**Figure 3. Complement exposure leads to progressive defect formation at the bacterial surface.**

(A) Larger-scale view of BL21(DE3) *E. coli* showing extensive damage on the bacterial membrane; AFM height (left) and phase (right) images. Darker areas in the phase images are regions that appear to have lost the outer membrane. (B) Sequence of AFM (height) images of BL21(DE3) *E. coli*, cropped from Movie EV1, showing MAC pores and larger (>50 nm wide) defects in the outer membrane. Data are shown in a 3D, tilted representation. Times are referenced with respect to the first recorded high-resolution image (0 min). (C) Brightfield (left) and SYTOX™ fluorescence (right) microscopy images of the bacteria in the AFM experiments, recorded 2 min before the AFM image, with red dashed ellipses indicating the location of the (SYTOX™ negative) cell imaged by AFM in (A). The dark disk is the AFM cantilever. (D) Brightfield (left) and SYTOX™ fluorescence (right) microscopy images recorded at the beginning of the AFM sequence displayed in (B), with red dashed ellipses indicating the location of the cell imaged by AFM. The dark paddle is the AFM cantilever. Color scale (see inset Fig. 1C): 8.5 deg (A, left), 350 nm (A, right), 30 nm (B). Scale bars: 100 nm (A), 50 nm (B), 1 μm (C, D).

and immediately before cell death as measured via inner membrane permeation. The swelling was accompanied by an increase in surface stiffness, as quantified by a change in effective Young's modulus (Fig. 4C,D). As was the case for the signatures of mechanical destabilization shown in Fig. 2, the swelling and surface stiffening were specific for complement-mediated killing; they were insignificant for bacteria that were lysed by the membrane-targeting peptide melittin (Appendix Fig. S13). The swelling was also confirmed by quantitative fluorescence microscopy (Middlemiss et al, 2024) over larger populations of cells that were exposed to ΔC8 serum ±C8C9, with optically measured mean cell widths of 1.17 μm without C8C9 and of 1.24 μm with C8C9 (Fig. 4E,F; Appendix Fig. S14).

## Discussion

The data presented here provide a unique perspective to complement-exposed bacterial cells in the stage between outer membrane perforation and subsequent inner membrane permeation and cell death. A surprising finding is the extent of outer membrane poration that the bacteria sustain before inner membrane permeation results, with not only tens to hundreds of MAC pores per cell (Figs. 1 and 2) but also propagating defects that vastly exceed the outer membrane damage done by the MACs alone (Fig. 3). Under the conditions of our experiments, the bacteria sustained such outer membrane damage for tens of minutes before inner membrane permeation was detected.

This temporal decoupling of outer membrane damage and inner membrane permeation is consistent with the hypothesis of an indirect route of inner membrane destabilization by complement (Doorduijn et al, 2019). Moreover, noting the observed swelling and surface stiffening immediately preceding cell death (Fig. 4), we conclude that this indirect route is a mechanical one.

In Gram-negative bacteria, the cell envelope experiences an outward osmotic (turgor) pressure, which is balanced by the cell wall with additional contribution from the outer membrane (Rojas et al, 2018; Sun et al, 2022). We therefore attribute the observed swelling and surface stiffening (Fig. 4) to the turgor pressure of the cell, pressing outwards against the cell envelope, which is increasingly destabilized due to complement exposure (Fig. 2). It leads to additional tension in the outer membrane, facilitating the formation of propagating defects in the MAC-porated outer membrane as observed in the high-resolution AFM images (Fig. 3), which in turn further destabilize the cell envelope. Such swelling also implies increased tension in the inner membrane, thereby increasing the likelihood of defect formation and facilitating cell entry of small molecules such as the nucleic acid dyes used for live/dead staining (Fig. 5).

In previous work, it was shown that such complement-induced inner membrane permeation does not directly translate into the release of larger biomolecules from the cytoplasm (Heesterbeek et al, 2019); accordingly, we expect that the turgor pressure may be reduced but does not collapse, such that swelling stops and that the surface stiffness reaches a plateau immediately after cell death, as we observe experimentally (Fig. 4, Appendix Figs. S10 and S11).

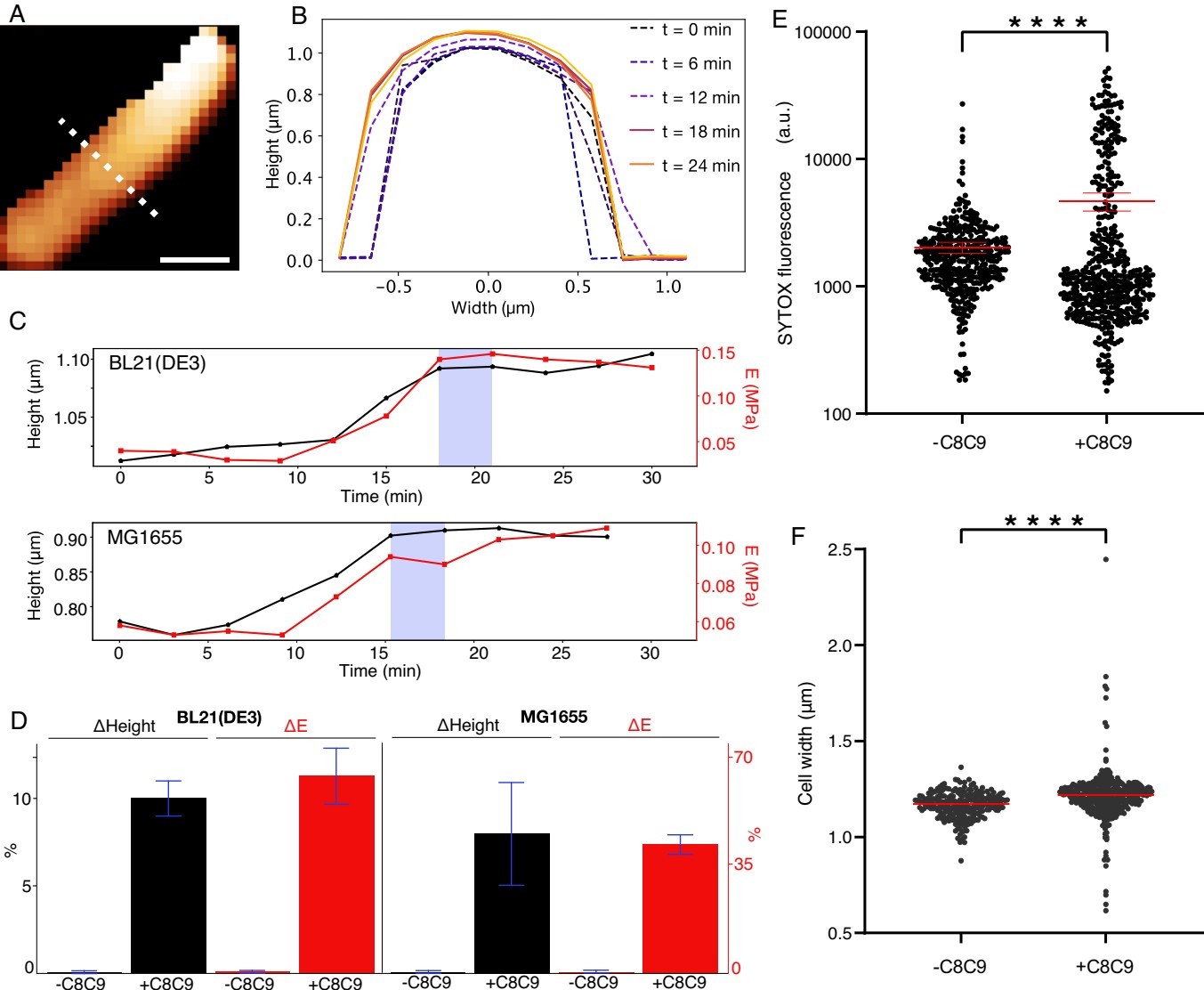

**Figure 4. Complement exposure leads to swelling and stiffening of bacteria before cell lysis.**

(A) AFM image of a BL21(DE3) cell, where each pixel represents the un-indented bacterial height. Color scale: 1 μm. Scale bar: 700 nm. (B) Height profiles, corresponding to the white dotted line in (A), plotted for different times. Dash profiles here indicate measurements for which the bacterium was still alive; solid lines indicate that the bacterium was dead (SYTOX™ positive). (C) Cell height (black, left axis) determined from line profiles as in (B), for a BL21(DE3) and for a MG1655 E. coli cell, and the effective Young's modulus (E, red, right axis) measured at the bacterial surface, as a function of time referenced to the start of the AFM measurements (0 min). Vertical, blue-shaded bands indicate the time at which the cell first stained SYTOX™ positive, indicative of inner membrane permeation, as shown by the fluorescence microscopy images (see Appendix Figs. S10–S12). (D) Quantification of the relative changes (mean, shown as bar height, ±1 standard deviation, shown as the blue error bars) in BL21(DE3) ($n = 3$ for height measurements, $n = 5$ for stiffness measurements) and MG1655 E. coli ($n = 3$) cells, comparing height and stiffness at the end with the beginning of the measurements as in (C), compared with a negative control (no C8 and C9). (E) Quantification of SYTOX™ fluorescence in MG1655 E. coli cells treated with or without C8 and C9 for 15 mins. $n = 396$ cells ($-$C8C9), 479 cells ($+$C8C9). Red bars represent the mean with 95% confidence interval. (F) Quantification of cell widths of E. coli MG1655 cells treated with or without C8 and C9 for 15 mins. $n = 221$ cells ($-$C8C9), 373 cells ($+$C8C9). Red bars represent median widths. P values were a result of an unpaired, two-tailed t test with Welch's correction, ****$P < 0.0001$ (E: $6.34 \times 10^{-11}$; F: $5.31 \times 10^{-6}$).

Considering the mechanics of the outer membrane, we can speculate that MAC pores act as mechanical defects that nucleate further structural and mechanical disruption of the outer membrane as observed in our data. In addition, outer membrane poration will also cause leakage of periplasmic contents: our data suggest that this affects maintenance of the cell wall and hence its mechanical integrity. In vivo, such effects on the cell wall will be reinforced further by the entry of lysozymes (Heesterbeek et al, 2021), which degrade the peptidoglycan (Wright and Levine, 1981).

As the defects propagate, they leave voids in the outer membrane (Fig. 3B; Appendix Figs. S7 and 8). This also implies the release of outer membrane material and entire or partial MAC assemblies from the bacterial surface—of note, released, soluble C5b–C9 assemblies (also known as sC5b-9 or sMACs) have been

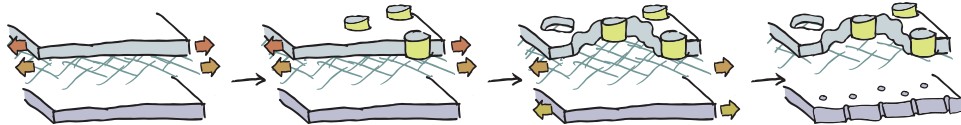

**Figure 5.  Schematic of the proposed mechanism for complement-induced bacterial killing.**

In Gram-negative bacteria, the outer membrane (top surface) and cell wall (peptidoglycan network; green) are both under tension (red and orange arrows) due to the turgor pressure. MAC (yellow) poration in the outer membrane compromises cell wall maintenance and overall ability of the cell envelope to resist the turgor pressure. This leads to and is further aggravated by extensive defect formation in the outer membrane; the cell swells. The inner membrane (bottom surface) is stretched in the process, resulting in increased permeability and hence cell lysis.

attributed roles in immune activation and bacterial resistance to complement (Barnum et al, 2020; Doorduijn et al, 2022).

In summary, these observations challenge the traditional understanding of complement-mediated killing as established since the first observations of lytic MAC pores six decades ago (Muller-Eberhard, 1986). We conclude that MAC pores are not lytic to the bacterial cell per se, but cause mechanical disruption of the outer membrane and cell wall and thereby of the two load-bearing elements of the Gram-negative bacterial cell envelope (Rojas et al, 2018; Sun et al, 2022). Hence, it is not membrane poration, but mechanical disruption that seals the bacteria's fate. The interaction between mechanical destabilization and other cell death pathways, for example, lipid exchange or stress responses (Doorduijn et al, 2019), will be an interesting avenue for further study.

This route to cell death raises new questions about the fundamental role of the outer membrane for the mechanical integrity of the cell wall, its role as a load-bearing element (Rojas et al, 2018; Sun et al, 2022; Fivenson et al, 2023) or in maintaining the periplasmic integrity and thereby protecting the cell wall.

# Methods

## Preparation of substrates for bacterial adhesion

Following procedures previously described elsewhere (Benn et al, 2019, 2021), 13-mm glass coverslips (VWR) were rinsed in a stream of milliQ (mQ) water, sonicated in 1–2% SDS at 37 kHz and 100% power in a Fisherbrand™ bath sonicator (Fisher Scientific) for 10 min, rinsed in mQ water, then ethanol, dried with nitrogen, plasma cleaned at 70% power for 2 min in a plasma cleaner in air (2.6 L Zepto, Diener Electronic); this cleaning procedure was then repeated two times. Clean coverslips were coated with poly-L-lysine (PLL) applying a 100 μl droplet of 0.01% poly-L-lysine (P4832, Sigma) for 5 min, rinsing in a stream of mQ water and drying with nitrogen; or coated with Vectabond®, by submersion in a 50:1 solution of Acetone:Vectabond® (Vector Laboratories, USA) for 5 min, followed by rinsing in mQ water and drying under nitrogen flow. Coated coverslips were glued to clean glass slides using biocompatible glue (Reprorubber thin pour, Flexbar).

## Sample preparation

BL21(DE3) or MG1655 *E. coli* were grown overnight in Lysogeny broth (LB, Lennox) at 37 °C in a shaking incubator, diluted into fresh LB and grown for 2.5–3 h. 0.5 to 1 ml of fresh culture was

spun at 5000 rpm for 2 min and resuspended in HEPES buffer (20 mM HEPES, 120 mM NaCl, pH 7.4). This was repeated 3 more times. In all, 100 μl of cells were applied to a coverslip, incubated for 5 min and washed three times with 1 ml of HEPES buffer. To reduce sequestering of proteins or peptides to the functionalized surface, bacteria were then washed into HEPES/BSA (20 mM HEPES, 120 mM NaCl, 2.5 mM MgCl$_2$, 0.1% BSA, pH 7.4) and incubated for 30 min at room temperature.

Bacteria were exposed to serum either in solution, or after immobilization onto glass. In both methods, bacteria were washed in HEPES/BSA buffer (20 mM HEPES, 120 mM NaCl, 2.5 mM MgCl$_2$, 0.1% BSA, pH 7.4) with 10% C5 (or C8) deficient human serum (Complement Technology, Texas, USA) and incubated at 37 °C for 20–30 min. Cells were then washed with HEPES buffer three times. If still in suspension, bacteria were then immobilized onto PLL/Vectabond-coated coverslips, as described above. For the application of C5-9 (or C8-9), immobilized bacteria were washed once with 1 ml of HEPES/BSA buffer, the droplet was removed and replaced by 100 μl HEPES/BSA with purified complement components (Complement Technology, Texas, USA). Concentrations were 0.04 mg/ml C5, 0.012 mg/ml C6 and C7, 0.015 mg/ml C8 and 0.007 to 0.07 mg/ml C9, thus ranging up to the approximate concentrations in serum (Sharp et al, 2016). Following exposure to C5-deficient serum, C5, 6, 7, 8, and 9 were either added together, or first C5, 6 and 7, followed by C8 and C9. Following exposure to C8 deficient serum, C8 and C9 were added together. AFM imaging was initiated 10–15 min after C9 incubation, with the first AFM images recorded shortly after.

For additional AFM measurements shown in Appendix Fig. S7, cells were prepared with minor deviations of the protocol given here, as detailed in Ref (Heesterbeek et al, 2019).

For AFM on cells targeted by the AMP melittin, exponential phase BL21(DE3) *E. coli* in HEPES buffer were immobilized onto PLL-coated coverslips. Before imaging by AFM, the droplet was made up to 150 μl. When ready for the application of melittin, 50 μl of the droplet was removed and replaced with HEPES buffer containing melittin for a final concentration of 5 μM. The droplet was then vigorously pipette mixed and AFM continued.

## Brightfield and fluorescence microscopy

An Andor Zyla 5.5 USB3 fluorescence camera on an Olympus IX 73 inverted optical microscope was used. Bacterial cell death was assessed using SYTOX™ green nucleic acid stain (S7020, Sigma). 0.3–1 μl of 5 mM stain was added to the sample, and images were acquired. If the stain bleached, more SYTOX™ was added.

For quantitative optical microscopy to assess pore formation via SYTOX staining and cell width measurements, overnight cultures of *E. coli* MG1655 were grown at 37 °C with orbital agitation at 175 rpm in LB (Oxoid). Overnight cultures were diluted 1:100 in fresh LB (Oxoid) and grown at 37 °C with orbital agitation at 175 rpm until mid-logarithmic phase was reached (OD600 = 0.50 +/− 0.05). Cells were resuspended in 200 μl HEPES/BSA buffer containing 10% (v/v) C8-deficient human serum (Complement Technology, Texas, USA), prior to being transferred to 2 ml Eppendorf tubes with 2x holes in the lid for aeration. Samples were incubated at 37 °C with shaking at 850 rpm in a benchtop heated shaker (Eppendorf) for 30 min. The bacteria were then washed three times with HEPES buffer before re-suspension in 200 μl HEPES/BSA buffer containing purified C8 and C9 complement proteins at final concentrations of 2.5% (v/v) and 7.5% (v/v), respectively. At this step either SYTOX (200 nM final) (Sigma) or the membrane dye FM5-95 (1 μg/ml final) (Invitrogen) was added. Samples were re-incubated at 30 °C for 7–15 min, see Figures for details, with shaking at 850 rpm before 0.5 μl cells were spotted onto multi-spot microscope slides (Hendley-Essex) containing a thin layer of 2% ultra-pure agarose (Invitrogen). Cells were air dried (~1 min) prior to the application of a high-precision coverslip (Thorlabs, 22 × 22 mm, thickness no. 1.5H).

A custom microscope was used for fluorescence Imaging as described previously (Middlemiss et al, 2024). Briefly, cells were illuminated using 488 nm (Vortran Stradus) or 561 nm (MPB Fibrelaser) lasers, a Nikon CFI Apochromat TIRF 100XC Oil objective and a 200 mm tube lens (Thorlabs TTL200). A Prime BSI sCMOS camera (Teledyne Photometrics) was used generating an apparent pixel size of 65 nm/pixel. Images were acquired using 50 ms (for SYTOX-stained cells) or 200 ms (for FM5-95-stained cells) exposures with an illumination power measured at the sample of 2.5 mW, generating a power density of 56.4 W/cm$^2$ over an illumination area of 1024 × 1024 pixels.

## AFM imaging

AC tapping mode AFM was performed in liquid, on a Nanowizard III AFM with an UltraSpeed head (Bruker AXS, CA, USA) with a FastScanD (Bruker AXS, CA, USA) cantilever (0.25 N m$^{-1}$ spring constant and 110 kHz resonant frequency in liquid). For AC mode, a drive frequency of ~100 kHz and amplitude of ~10 nm were used, corresponding to an ~40% drop from the free amplitude. For PeakForce Tapping, the drive frequency was 8 kHz. Images were acquired at 256 × 256 or 512 × 512 pixels with an aspect ratio of 1:1. Images size varied between 300 and 500 nm for surface scans. Whole-cell images were collected with 2–10 μm scans.

Fast force spectroscopy, PeakForce Tapping AFM was performed in liquid on a Resolve microscope (Bruker, SXS, CA, USA) with PF HIRS-F-B cantilevers (Bruker AXS, CA, USA) with 0.12 N/m spring constant, and 1 nm nominal tip radius. The cell surface was imaged at 0.5 Hz, maximum force of 150 to 200 pN, PeakForce amplitude of 20 nm, low-pass bandwidth of 10 Hz, Peak Force frequency of 2 kHz at a Z range of 3 μm. Images were acquired at 128 × 128, or 256 × 256 pixels with 1:1 aspect ratio. Additional PeakForce Tapping measurements (Appendix Fig. S7) were performed in liquid on a Fastscan Dimension system (Bruker, SXS, CA, USA) with FastScanD probes as above, with 8 kHz

PeakForce frequency; for the data recorded on this system, cells were prepared with minor deviations of the protocol given here, as detailed in Ref (Heesterbeek et al, 2019).

For mechanical characterization by recording force-vs-indentation curves, we used quantitative imaging (QI) mode AFM. This was performed in liquid on a Nanowizard III AFM with an UltraSpeed head (Bruker AXS, CA, USA) with PFQNM-LC-CAL (Bruker AXS, CA, USA) pre-calibrated cantilever (0.05–0.08 N/m spring constant). Images were collected as 32 × 32, 12 × 12 pixels with 0.2 nN set-point, 1200 nm ramp height, 1.5 × 1.5 μm$^2$ typical scan size, 2 μm/s scan speed, aspect ratio 1:1.

## Analysis of AFM images

Processing of AFM images was performed using the Gwyddion 2.52 (http://gwyddion.net/) pygwy module (Nečas and Klapetek, 2012) in python, using a script adapted from open-source AFM image analysis software (Beton et al, 2021) (source code available at https://github.com/AFM-SPM/TopoStats/blob/bacteria/bacterial_image_processing.py).
Briefly, images were processed depending on type and size. Large phase and height images were masked to ignore high pixels, then plane leveled. Small phase images were processed by aligning rows with a 2nd-order polynomial fit and applying a 1-pixel Gaussian filter. For small height images, sequentially, the flatten base algorithm, a 2nd-order polynomial fit and a 1-pixel Gaussian filter were applied. For RMS values, height images were cropped to include only well imaged regions, rows were then aligned with a 2nd-order polynomial fit and the RMS calculated.

For high-resolution whole-cell images, processed, small scans were overlaid to cover most of the cell surface. Image alignment was achieved by first finding approximate scan locations in the JPK data processing software. To correct for drift, images were then manually overlaid using FIJI-ImageJ (Schindelin et al, 2012). Brightfield and fluorescence images were also overlaid and cropped in FIJI-ImageJ (Schindelin et al, 2012).

For quantification of MAC deposition, the coordinates of individual MACs were picked manually with the multi-point selection tool in FIJI-ImageJ (Schindelin et al, 2012). The total area of each cell surface imaged at a suitable resolution was found by manual selection. All quantifications were compared to the same number of random coordinates as MACs generated 5 times, over the same area, for each bacterium, in MATLAB.

To assess the distribution of MACs across the surface, MACs were binned into five regions from midpoint to pole and the number in each section was normalized to the region area (Appendix Fig. S2).

For clustering analysis, each MAC point was dilated in FIJI-ImageJ (Schindelin et al, 2012) to join close points as continuous shapes. The skeletonize function was applied, and the longest branch of each skeleton was found (Appendix Fig. S3). MAC coordinates were also used to calculate the nearest neighbor distances and the pair distribution function $g(r)$ in MATLAB.

Statistics were performed either in MATLAB (Mathworks) or Origin (OriginLab, MA, USA) and graphs were plotted in Origin (OriginLab, MA, USA). Statistics are from a paired two-sided Student's *t* tests, a one-way ANOVAs with a Tukey's *t* test or a $\chi^2$ test. Unless otherwise specified, error bars are standard deviations, and center lines are means. Codes are available at https://github.com/hoogenboom-lab/image-analysis.

## Analysis of AFM mechanical data

AFM images collected in QI mode (Fig. 4; Appendix Figs. S10–S13) were analyzed using Nanowizard SPM software (Bruker AXS, CA, USA). Raw data were batch-processed; all force curves of a map were analyzed together so that processing operations could be combined and executed automatically for a large number of force curves. Data were calibrated by the known spring constant $k$ (provided by the manufacturer for the pre-calibrated cantilevers as used to acquire these data) and the cantilever deflection sensitivity as determined from thermal noise data using the known $k$. A constant baseline was calculated by averaging the last 10% of the curve (i.e., in the region far from the surface, where there is no force between surface and tip); this value was then subtracted from the whole curve. The contact point was first estimated by detecting where the force curve first deviated from the zero-force line. This value was then used to adjust the horizontal offset (i.e., the zero of the horizontal axis was set to this value). The tip-vertical position was calculated to correct the height for the bending of the cantilever. Finally, to determine an effective Young's modulus as a measure of the surface mechanics, a Hertz indentation model was fitted to the data, with tip shape assumed to locally resemble a sphere and assuming a Poisson ratio of 0.5.

## Data availability

Source data for the main figures in this paper are freely available at the University College London research data repository, https://doi.org/10.5522/04/26268520.v1.

The source data of this paper are collected in the following database record: biostudies:S-SCDT-10_1038-S44318-024-00266-3.

## Peer review information

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

## Acknowledgements

The authors acknowledge Richard Thorogate (University College London) for technical support, and Isabel Bennett, Edward Parsons (University College London), Dani Heesterbeek, Bart Bardoel and Suzan Rooijakkers (University Medical Centre Utrecht) for discussions and for support with/advice on sample preparation, and Charles Clifford and Caterina Minelli (National

Physical Laboratory) for comments on the manuscript. Research reported in this publication was funded by the United Kingdom Research and Innovation Biotechnology and Biological Sciences Research Council (BB/R000042/1 and BB/X001547/1 to BWH), Engineering and Physical Sciences Research Council (EP/N509577/1 to GB and BWH; EP/K031953/1 for equipment via the Interdisciplinary Research Centre in Early-Warning Sensing Systems for Infectious Diseases), Medical Research Council (MR/W00738X/1 and MR/R024871/1 to ALBP; MR/R000328/1 and MR/V009702/1 to BWH), and the UK Department for Science, Innovation and Technology (to CB). Work in the Holden lab was supported by a Wellcome Trust & Royal Society Sir Henry Dale Fellowship (206670/Z/17/Z).

## Author contributions

**Georgina Benn**: Conceptualization; Data curation; Formal analysis; Investigation; Visualization; Methodology; Writing—original draft; Writing—review and editing. **Christian Bortolini**: Conceptualization; Data curation; Formal analysis; Investigation; Visualization; Methodology; Writing—original draft; Writing—review and editing. **David M Roberts**: Conceptualization; Data curation; Formal analysis; Investigation; Visualization; Methodology; Writing—review and editing. **Alice L B Pyne**: Conceptualization; Data curation; Formal analysis; Validation; Investigation; Visualization; Methodology; Writing—review and editing. **Séamus Holden**: Conceptualization; Funding acquisition; Methodology; Project administration; Writing—review and editing. **Bart W Hoogenboom**: Conceptualization; Data curation; Supervision; Funding acquisition; Investigation; Methodology; Writing—original draft; Project administration; Writing—review and editing.

Source data underlying figure panels in this paper may have individual authorship assigned. Where available, figure panel/source data authorship is listed in the following database record: biostudies:S-SCDT-10_1038-S44318-024-00266-3.

## Disclosure and competing interests statement

The authors declare no competing interests. BWH holds an executive position at AFM manufacturer Nanosurf, but Nanosurf did not play any role in the design or execution of this study.

