## [Peer Review File · The EMBO Journal]

Complement-mediated killing of *Escherichia coli* by mechanical destabilization of the cell envelope

Georgina Benn, Christian Bortolini, David Roberts, Alice Pyne, Seamus Holden, and Bart Hoogenboom

Corresponding author(s): Bart Hoogenboom (B.hoogenboom@ucl.ac.uk)

Review Timeline:

Submission Date:	4th Mar 24
Editorial Decision:	10th Apr 24
Revision Received:	3rd Jul 24
Editorial Decision:	1st Sep 24
Revision Received:	14th Sep 24
Accepted:	29th Sep 24

Editor: Ieva Gailite

Transaction Report:

Dear Dr. Hoogenboom,

Thank you for submitting your manuscript for consideration by the EMBO Journal. We have now received comments from three reviewers, which are included below for your information.

As you will see from the reports, all reviewers find the study per se of interest, while also indicating its descriptive nature. The reviewers ask for a better quantification of the findings, while also indicating that expanding the analysis to other Gram-negative bacteria and to bacteria with altered outer membrane mechanical properties would enhance the impact of the study.

Based on the interest expressed in the reports, I invite you to address these concerns in a revised version of the manuscript. I think it would be helpful to discuss the revision in more detail via email or phone/videoconferencing - please let me know which option you prefer. I should also add that it is The EMBO Journal policy to allow only a single major round of revision and that it is therefore important to resolve the main concerns at this stage.

We generally allow three months as standard revision time, which can be extended to six months in the case of major revisions. As a matter of policy, competing manuscripts published during this period will not negatively impact on our assessment of the conceptual advance presented by your study. However, please contact me as soon as possible upon publication of any related work to discuss the appropriate course of action. Should you foresee a problem in meeting this deadline, please let us know in advance to discuss an extension.

When preparing your letter of response to the referees' comments, please bear in mind that this will form part of the Review Process File and will therefore be available online to the community. For more details on our Transparent Editorial Process, please visit our website: <https://www.embopress.org/page/journal/14602075/authorguide#transparentprocess>. Please also see the attached instructions for further guidelines on preparation of the revised manuscript.

Please feel free to contact me if you have any further questions regarding the revision. Thank you for the opportunity to consider your work for publication. I look forward to discussing your revision.

With best regards,

Ieva

Ieva Gailite, PhD
Senior Scientific Editor
The EMBO Journal
Meyerohofstrasse 1
D-69117 Heidelberg
Tel: +4962218891309
i.gailite@embojournal.org

- a point-by-point response to the referees' comments, with a detailed description of the changes made (as a word file).
- a word file of the manuscript text.
- individual production quality figure files (one file per figure)
- a complete author checklist, which you can download from our author guidelines

(<https://www.embopress.org/page/journal/14602075/authorguide>).

- Expanded View files (replacing Supplementary Information)

We realize that it is difficult to revise to a specific deadline. In the interest of protecting the conceptual advance provided by the work, we recommend a revision within 3 months (9th Jul 2024). Please discuss the revision progress ahead of this time with the editor if you require more time to complete the revisions. Use the link below to submit your revision:

Referee #1:

Manuscript by Benn and colleagues explores killing of bacteria by membrane attack complex of complement. It is well established that membrane attack complex forms pores at the surface of Gram-negative bacteria and that this is the reason for cell death (lysis). However, it is less clear how cell death proceeds after formation of pores on the surface. In this manuscript, authors studied changes in the shape and mechanical properties of cell induced by membrane attack complex pores at the membrane surface. The results show that formation of pores cause large-scale destruction of the outer membrane, swelling and stiffening of the bacterial cell and, finally, inner membrane permeation. This study systematically explores steps subsequent to pore formation at the bacteria surface and provide important data and concepts related to complement-induced bacterial killing. The following comments may further improve the manuscript.

It is claimed on two occasions (in the abstract and discussion sections) that this newly described route to bacterial cell death could be exploited by novel antibiotic treatments. This is not addressed in the manuscript or explored in depth and should, therefore, be omitted.

In the third line of the introduction, a non-trivial mechanical behavior is mentioned. What is this? This should probably be phrased differently.

AFM imaging of MAC pores is convincing (Figure 1 A,C). Is there any way to know whether the circular assemblies are functional? How do we know that the pores formed on the surface of the bacterial cell shown in Figure 1A are functional, i.e. penetrate the outer membrane? Can the height of the pores be indicative of inserted vs non-inserted pores, as was shown for cholesterol-dependent cytolysins? It is claimed that MACs protrude several nanometers from the surface. Can authors be more exact and quantitative here (page 4, second paragraph)?

It is mentioned on several occasions that outer membrane contains tens to hundreds of MAC pores (i.e. page 4, third paragraph). Can authors be more quantitative be here? For examples presented in this manuscript the number of pores per cell could be counted probably and specified in the text. For example, on figure 1A several tens of pores are shown, but not hundreds.

Black holes, gaps in the membranes, should be labeled in Figure 2 for clarity.

Data are shown for E. coli strains BL21 and MG1655. In order to show that cell death induced by MAC pores proceeds in a similar manner in Gram-negative bacteria, some more examples could be shown. It would be nice to show this in other Gram-negative bacteria. But also in E. coli mutants that have altered mechanical properties. Or perform experiments on cells pretreated with lysozyme, as this would weaken the stiffness of the cell envelope and would result in different kinetics of the events described in the manuscript.

Strains BL21 and BL21(DE3) are mentioned in the text on several occasions. Which was used? Please clarify.

Legend to Supplementary Fig. 3 should contain description of all panels (check E and F panels).

It is not clear how the stability of the outer membrane can be inferred from the images presented in Supp Figure 4. Please clarify

and provide more explanations in the text.

Supp Figure 5 shows disruption of cell envelope, but this is not clearly presented. The figure clearly shows feature of the outer surface with many MAC pores that gradually disappear in time. How do we see major disruption? Could some more examples be shown here?

What is a violet rectangle in Supp Figures 10-12, please specify in the legend to figures? Why is no data shown for height for the cells presented in D and E in Supp Figure 10?

Cells presented in Supp Figure 14B in the presence of C8C9 (right panel of 14B) do not seem swelled (as is claimed on page 8, third paragraph). They are shorter than cells in the absence of C8C9, but rounder and not wider. Please clarify.

Referee #2:

The manuscript submitted by Benn et al concerns an essential mechanism in innate immunity, how insertion of the membrane attack complex in the outer membrane of gram-negative bacteria leads to cell killing. The major result of the study is a model for how MAC insertion propagates through perforation of the outer membrane to swelling and formation of holes in the inner membrane

The manuscript is well written and easy to follow, also for the non-expert. The conclusions made based on the data appears sound and justified. The technical quality is appropriate. This is mainly an observational study, an attempt to reach detailed mechanistic insight is not dominating.

Major issues

The study is highly dominated by time-dependent AFM examination of E coli cells after induction of MAC assembly. It would increase the general value of the study if a few other gram negative bacteria were investigated as well to demonstrate that the model applies broadly to gram negative organisms. There may also be E coli strains with mutations that change the properties of the outer membrane that could be included.

Minor issues.

It would help the reader if a sketch of the two membranes and the MAC drawn to scale was presented in Figure 1 to introduce the system.

Perhaps trivial, but there appears not to be an AFM recording of untreated and C5b-C7 treated cells for direct comparison with the beautiful pictures of MAC perforated E coli presented throughout the manuscript.

Abstract, first line. Serum is mentioned, but serum is an in vitro reagent. Change to the in vivo compartments where bacteria are killed by MAC.

Abstract. Last sentence is unjustified, there is not a single suggestion in the manuscript on how to take advantage of this knowledge for creating new antibiotics. Such discussion should be added or the sentence removed from the abstract Introduction. "Nonetheless, previous research has shown that bacteria can be lysed by C9 when the only available C5b-C8 is bound to the outer membrane, meaning that lysis must be the result of MACs formed only in the outer membrane and not in the inner membrane"

Meaning of sentence is unclear, revise.

Materials and methods. In a few places, a space should be inserted between the number and the unit

Referee #3:

Review of Benn - Complement-mediated killing of bacteria by mechanical destabilization of the cell envelope

This manuscript describes phenomenology associated with the series of events that lead to cell death after exposure to complement, as assayed via various modalities of AFM. The authors find that within minutes after MACs form, membrane defects begin to form and spread, followed by widening of the cell, stiffening of the cell, fluidization or softening of the outer membrane, and finally cell death.

Many of the micrographs are quite remarkable; in and of themselves they provide visualization and some intuition for what happens upon treatment with complement, which is of great biological and medical interest. The main conceptual result of the paper is that undermining the mechanical integrity of the outer membrane is the key step (or is a key step), and is the ultimate step, in cell lysis. The key piece of data is that AFM images get blurred and corrugated in the time window leading up to cell lysis.

Major Comments:

-This is a valuable study. It provides a new phenomenological window into complement-mediated killing. This will be of particular interest to medical microbiologists interested in Gram-negative pathogens. The micrographs will provide valuable insights to this community.

-The central claim seems a little strong to me. What the authors have done is show, upon complement treatment, a temporal correlation between cell lysis and outer membrane structure and mechanical properties. It would be surprising if these latter effects were not contributing to lysis, but because of the narrow focus of the study on the outer membrane it seems they are de facto interpreting lysis as exclusively due to mechanical destabilization of the outer membrane. In principle, this could be true, but it would be surprising (to me) if there were not other harmful physiological changes occurring simultaneously with damage to the outer membrane (e.g. changes to peptidoglycan structure, free radicals, protein stress, membrane depolarization, PG and OM synthesis, etc.) that also contribute to cell death via a mechanism that may or may not involve the outer membrane. If they measured these and found one of them to also correlate with lysis progression, it is unclear to me whether or not they would still say that complement "kills bacteria by mechanical destabilization." This is important because the time scales of lysis are quite long (tens of minutes in many cases). At these time's hard to imagine that there is not a global dysbiosis occurring in the cell.

This is primarily as an interpretation/communication issue rather than a critique of the results. In my opinion, the manuscript does not quite show that "mechanical destabilization kills bacteria when they are subjected to complement" it shows that "the outer membrane becomes mechanically destabilized when bacteria are subjected to complement, which contributes to lysis." If the authors really wanted to show that outer membrane strength is the key limiting factor in complement-mediated killing, they would have to predictively tune outer membrane strength (through mutations to outer membrane composition) or the ability of complement to destabilize it (without, somehow, inhibiting its ability to form complexes), and then show that this meaningfully altered the ability of complement to kill cells.

-As it stands, Figures 1-3 are qualitative and may or may not represent the behavior of an average cell. For me, it was somewhat difficult to piece together what would be happening to such a cell. For example, Figures 2 and S5 are the key figures showing that blurring of the AFM precedes lysis, but they occur on vastly different time scales, which is interesting but also confusing. There are no n numbers, statistics, or quantifications provided for these measurements.

-Similarly, it would be very helpful to have the distribution of lysis times as assayed via fluorescence microscopy and any such temporal information about the temporal dynamics of blurring of the AFM since this correlation is the central point of the paper.

-In line with the points above: the authors argue that because cells "could show no staining for inner membrane permeation" upon complement treatment, that outer membrane perforation is not sufficient to lyse cells. However, in figures 2A (and maybe 1B?), there appear to be lysed cells at 0 minutes after complement treatment. It is unclear what this means for the model although it is very interesting and, once again, begs for quantification of lysis times (at least) and ideally some explanation of what is presumably a broad distribution of lysis times.

Minor Comments

-In Fig. 1 can you show the corresponding image of an untreated control cell so that the reader can directly compare without having to look for it in another reference. Can you also add the n number. Quantifying the blurriness and showing the time courses for more than 1 cell would be helpful.

-In Fig. 2, can you show an image from the untreated control for reference. Can you also add the n number. Quantifying the blurriness and showing the time courses for more than 1 cell would be helpful.

-In Fig. 3B, is it possible to show the untreated (pre-treatment) control from the same area? Can you also add the n number and note how long after complement treatment 0 minutes corresponds to? Can you also label the MACs and the defects on the image, for a while I assumed the defects were the MACs.

-Fig. S5 seems to belong in the main text.

-In Fig. 4D, it is not useful to normalize to zero since now there is no information in the control bars (and no bars). If it's internally controlled in each experiment then it is more standard to normalize to 1. 5-10% difference for these variables is relatively large.

Response to Reviewers

Referee #1:

Manuscript by Benn and colleagues explores killing of bacteria by membrane attack complex of complement. It is well established that membrane attack complex forms pores at the surface of Gram-negative bacteria and that this is the reason for cell death (lysis). However, it is less clear how cell death proceeds after formation of pores on the surface. In this manuscript, authors studied changes in the shape and mechanical properties of cell induced by membrane attack complex pores at the membrane surface. The results show that formation of pores cause large-scale destruction of the outer membrane, swelling and stiffening of the bacterial cell and, finally, inner membrane permeation. This study systematically explores steps subsequent to pore formation at the bacteria surface and provide important data and concepts related to complement-induced bacterial killing. The following comments may further improve the manuscript.

We thank the reviewer for this positive and constructive feedback to our manuscript. In the following, we address this feedback point by point, where appropriately highlighting how we have revised the manuscript accordingly.

It is claimed on two occasions (in the abstract and discussion sections) that this newly described route to bacterial cell death could be exploited by novel antibiotic treatments. This is not addressed in the manuscript or explored in depth and should, therefore, be omitted.

This is indeed an extrapolation rather than something that we have demonstrated or explored in depth. Following the reviewer's recommendation, we have removed the speculative claims that this route to cell death could be exploited by novel antibiotic treatments.

In the third line of the introduction, a non-trivial mechanical behavior is mentioned. What is this? This should probably be phrased differently.

We have clarified this by different, and in our view, improved phrasing.

AFM imaging of MAC pores is convincing (Figure 1 A,C). Is there any way to know whether the circular assemblies are functional? How do we know that the pores formed on the surface of the bacterial cell shown in Figure 1A are functional, i.e. penetrate the outer membrane?

This was validated by our previous work as referred to in the introduction, “this detection of MACs on single cells correlated with outer membrane perforation and cell death in population assays [Heesterbeek et al., EMBO J 2019; Doorduyn et al., PLoS Path 2020]”. In those studies, we showed that outer membrane perforation and cell death were observed under conditions where AFM observed these circular assemblies. Similarly, when population (flow cytometry) assays indicated weakly bound and non-inserted MACs, AFM could not detect any of these assemblies, presumably since the non-functional MACs that do not perforate the membrane are too mobile or poorly adhered to the membrane. We conclude, based on these studies, that when MACs are seen by AFM, they must be functional, outer membrane-permeating pores.

Can the height of the pores be indicative of inserted vs non-inserted pores, as was shown for cholesterol-dependent cytolysins? It is claimed that MACs protrude several nanometers from the surface. Can authors be more exact and quantitative here (page 5, second paragraph)?

Cholesterol dependent cytolysins (CDCs) and MACs are part of the same protein superfamily but insert in different ways: CDCs tend to oligomerize at the surface, then insert together, giving a detectable drop in ring height. However, C9 monomers insert as they oligomerize into a ring [Parsons et al., Nat Commun 2019]. As a result, if there were a height difference, it would be hard to detect as it will be continuous, changing as each monomer inserts, rather than a step change in height like that observed for CDCs (by others and by ourselves [Leung et al. eLife 2014]). Moreover, C9 is more closely related to perforin than to CDCs, and perforin does not show a collapse in height upon transiting from a non-inserted to an inserted pore state [Leung et al. Nat Nanotechnol 2017]. Finally, it has previously been shown that MACs detected by AFM on *E. coli* are truly inserted, as mentioned in the previous comment and verified by trypsin incubation which does not dislodge these assemblies [Doorduyn et al. PLOS Path 2020].

We have now provided a more quantitative value (“MACs appeared to protrude approximately 3 nanometres from the surface...”, page 4 of manuscript with changes tracked), although these are still estimates. MACs are often densely packed and the cell surface does not provide a clear reference of where the membrane surface is, due to the presence of complement proteins (other than MAC) following treatment with serum. As a consequence, the protrusion above the background is not a generally accurate measure of MAC height above the membrane.

It is mentioned on several occasions that outer membrane contains tens to hundreds of MAC pores (i.e. page 4, third paragraph). Can authors be more quantitative be here? For examples presented in this manuscript the number of pores per cell could be counted probably and specified in the text. For example, on figure 1A several tens of pores are shown, but not hundreds.

We apologize for having caused confusion here and acknowledge the reviewer for this question: It is due to a typo in our original submission. We had already quantified the number of MACs per μm^2 as shown in Supplementary Figure 1, but the manuscript erroneously referred to the main Fig. 1. We have now corrected this and put the units in the main text (page 4, bottom, in manuscript with changed tracked). “despite their outer membrane being perforated by tens to hundreds of MAC pores per μm^2 (see Appendix Figure S1).”

Black holes, gaps in the membranes, should be labeled in Figure 2 for clarity.

Examples of these have now been labelled with white dotted lines in the left panel of Fig. 2B, where imaging is still clear enough to identify holes.

Data are shown for E. coli strains BL21 and MG1655. In order to show that cell death induced by MAC pores proceeds in a similar manner in Gram-negative bacteria, some more examples could be shown. It would be nice to show this in other Gram-negative bacteria. But also in E. coli mutants that have altered mechanical properties. Or perform experiments on cells pretreated with lysozyme, as this would weaken the stiffness of the cell envelope and would result in different kinetics of the events described in the manuscript.

These are very reasonable recommendations, but we are limited by the very low throughput of these single-cell experiments.

In our experiments, the cells are pre-incubated with serum components at a dose that leads to killing of cells on the time scale of tens of minutes to several hours, to enable us to initiate the AFM experiments and to next acquire data both before and after cell death. However, since we can only image one cell at the time and find a high variability in time to lysis, it is very much the question of being at the right place (cell) at the right time to capture data of a cell just before and just after cell death, as is needed here. Although we have optimized protocols (for *E. coli*) to the best of our abilities, this implies a low success rate in catching a cell while dying. When we succeed in doing so,

the results are consistent and repeatable as demonstrated in the manuscript, but it greatly limits our scope to expand the results to other mutants, strains or species within a practical time frame.

As for other Gram-negative bacteria, we note the general acceptance of *E. coli* as a model for Gram-negative bacteria and we expect our observation to extend to other complement-sensitive Gram-negative bacteria. However, as correctly implied by the reviewer, we have not demonstrated that this is really the case. We have therefore modified the title to make it clear upfront that our study relates to *E. coli* and not – or at least not in a proven way – to Gram-negative bacteria in general.

For comparing killing kinetics of different mutants, an additional difficulty is that one can only properly compare results in such experiments if the amount of MAC formation and poration is the same. We have in fact attempted to modulate mechanical stress, for example by osmotic pressure or EDTA treatment, but have so far failed to properly control for the amount of MAC deposition and pore formation in such experiments, leaving us with uninterpretable results.

Pretreatment with lysozymes is not expected to weaken the bacteria, since prior to outer membrane perforation by MAC pores, lysozymes cannot cross the outer membrane and access its target in the cell wall.

Strains BL21 and BL21(DE3) are mentioned in the text on several occasions. Which was used? Please clarify.

It was BL21(DE3) as now also clarified throughout the text.

Legend to Supplementary Fig. 3 should contain description of all panels (check E and F panels).

Thanks for highlighting this. There were two typos in the labels as given in the caption of this figure, which have now been corrected.

It is not clear how the stability of the outer membrane can be inferred from the images presented in Supp Figure 4. Please clarify and provide more explanations in the text.

Since atomic force microscopy is a surface scanning technique, it requires a stable, fairly static sample. Streaks in images and abrupt, diffuse roughening of a surface (as seen in Figure 2 and

supplementary Figures 4 and 5) imaged by AFM is indicative of detachment from an underlying support structure and/or readier mechanical dislodgement by to the scanning AFM tip. Since image quality is still good on other cells and the general shape of cells can still be detected (Supplementary Figure 4), we conclude this is not a perturbation of the whole cell, but instability of the outermost layer. We see that this was not obvious, so have made this clearer in the caption of Supplementary Figure 4 and the text (bottom of p. 6 in manuscript with changes tracked).

Supp Figure 5 shows disruption of cell envelope, but this is not clearly presented. The figure clearly shows feature of the outer surface with many MAC pores that gradually disappear in time. How do we see major disruption? Could some more examples be shown here?

As discussed above, the disappearance of MACs in images and an increase in roughening (measured by RMS) is indicative of outer membrane disruption. We have explained this more explicitly in the text and legends, and included more data in Supplementary Fig. 5 as asked for by the reviewer.

What is a violet rectangle in Supp Figures 10-12, please specify in the legend to figures?

This is now explained in the figure captions. Vertical, blue-shaded bands indicate the time at which the cell first stained SYTOX™ positive, indicative of inner membrane permeation, as shown by the fluorescence microscopy images.

Why is no data shown for height for the cells presented in D and E in Supp Figure 10?

For these cells, data had only been recorded on top of the cell, and not on the surrounding glass substrate. As a consequence, no reference heights were available for the cells in D and E, and therefore no cell height measurements (as included in A, B, C). We felt that this was no reason to exclude the datasets in D and E, since the stiffness and fluorescence data for these cells are no less valid.

Cells presented in Supp Figure 14B in the presence of C8C9 (right panel of 14B) do not seem swelled (as is claimed on page 8, third paragraph). They are shorter than cells in the absence of C8C9, but rounder and not wider. Please clarify.

We have included 2 additional representative membrane-stained images from both the -C8C9 and +C8C9 treatment conditions for Supplementary Fig 14B, thereby better illustrating the populations of cells (e.g. shorter and longer cells in both +/- C8C9 conditions). The swelling (measured in this case as a change in width) upon addition of C8C9 is relatively subtle, and hard to see by eye, which is why we also quantified the width change over hundreds of cells, confirming a statistically significant increase in width +C8C9 (Fig 4F).

Referee #2:

The manuscript submitted by Benn et al concerns an essential mechanism in innate immunity, how insertion of the membrane attack complex in the outer membrane of gram-negative bacteria leads to cell killing. The major result of the study is a model for how MAC insertion propagates through perforation of the outer membrane to swelling and formation of holes in the inner membrane. The manuscript is well written and easy to follow, also for the non-expert. The conclusions made based on the data appears sound and justified. The technical quality is appropriate. This is mainly an observational study, an attempt to reach detailed mechanistic insight is not dominating.

We thank the reviewer for these kind comments on our manuscript and for the constructive feedback below.

Major issues

The study is highly dominated by time-dependent AFM examination of E coli cells after induction of MAC assembly. It would increase the general value of the study if a few other gram negative bacteria were investigated as well to demonstrate that the model applies broadly to gram negative organisms. There may also be E coli strains with mutations that change the properties of the outer membrane that could be included.

These are very reasonable recommendations, but we are limited by the very low throughput of these single-cell experiments.

In our experiments, the cells are pre-incubated with serum components at a dose that leads to killing of cells on the time scale of tens of minutes to several hours, to enable us to initiate the AFM experiments and to next acquire data both before and after cell death. However, since we can only image one cell at the time and find a high variability in time to lysis, it is very much the question of being at the right place (cell) at the right time to capture data of a cell just before and just after cell death, as needed here. Although we have optimized protocols (for *E. coli*) to the best of our abilities, this implies a low success rate in catching a cell while dying. When we succeed in doing so, the results are consistent and repeatable as demonstrated in the manuscript, but it greatly limits our scope to expand the results to other mutants, strains or species within a practical time frame.

As for other Gram-negative bacteria, we note the general acceptance of *E. coli* as a model for Gram-negative bacteria and we expect our observation to extend to other complement-sensitive, Gram-negative bacteria. However, as correctly implied by the reviewer, we have not demonstrated that this is really the case. We have therefore modified the title to make it clear upfront that our study relates to *E. coli* and not – or at least not in a proven way – to Gram-negative bacteria in general.

For comparing killing kinetics of different mutants, an additional difficulty is that one can only properly compare results in such experiments if the amount of MAC formation/poration is the same. We have in fact attempted to modulate mechanical stress, for example by osmotic pressure or EDTA treatment, but have so far failed to properly control for the amount of MAC deposition and pore formation in such experiments, leaving us with uninterpretable results.

Minor issues.

It would help the reader if a sketch of the two membranes and the MAC drawn to scale was presented in Figure 1 to introduce the system.

This has now been included as the new Fig. 1A.

*Perhaps trivial, but there appears not to be an AFM recording of untreated and C5b-C7 treated cells for direct comparison with the beautiful pictures of MAC perforated *E. coli* presented throughout the manuscript.*

In the new Fig. 1B, we have now included comparable images of the surface of an untreated cell and of a cell treated with complement up to C5, in addition to the previously included images of cells treated with complement up to C9, showing MACs.

Abstract, first line. Serum is mentioned, but serum is an in vitro reagent. Change to the in vivo compartments where bacteria are killed by MAC.

We have changed this to “in the blood” (instead of “in serum”).

Abstract. Last sentence is unjustified, there is not a single suggestion in the manuscript on how to take advantage of this knowledge for creating new antibiotics. Such discussion should be added or the sentence removed from the abstract.

This is indeed an extrapolation rather than something that we have demonstrated or explored in depth. Following the reviewer's recommendation, we have removed the speculative claims that this route to cell death could be exploited by novel antibiotic treatments.

Introduction. "Nonetheless, previous research has shown that bacteria can be lysed by C9 when the only available C5b-C8 is bound to the outer membrane, meaning that lysis must be the result of MACs formed only in the outer membrane and not in the inner membrane". Meaning of sentence is unclear, revise.

This has now been clarified by revision in the 2nd paragraph of the introduction.

Materials and methods. In a few places, a space should be inserted between the number and the unit.

At several occurrences, such spaces have been inserted in this revised manuscript.

Referee #3:

Review of Benn - Complement-mediated killing of bacteria by mechanical destabilization of the cell envelope

This manuscript describes phenomenology associated with the series of events that lead to cell death after exposure to complement, as assayed via various modalities of AFM. The authors find that within minutes after MACs form, membrane defects begin to form and spread, followed by widening of the cell, stiffening of the cell, fluidization or softening of the outer membrane, and finally cell death.

Many of the micrographs are quite remarkable; in and of themselves they provide visualization and some intuition for what happens upon treatment with complement, which is of great biological and medical interest. The main conceptual result of the paper is that undermining the mechanical integrity of the outer membrane is the key step (or is a key step), and is the ultimate step, in cell lysis. The key piece of data is that AFM images get blurred and corrugated in the time window leading up to cell lysis.

We thank the reviewer for these kind comments on our manuscript and for the constructive feedback below.

Major Comments:

-This is a valuable study. It provides a new phenomenological window into complement-mediated killing. This will be of particular interest to medical microbiologists interested in Gram-negative pathogens. The micrographs will provide valuable insights to this community.

We appreciate the overall positive assessment of our manuscript here.

-The central claim seems a little strong to me. What the authors have done is show, upon complement treatment, a temporal correlation between cell lysis and outer membrane structure and mechanical properties. It would be surprising if these latter effects were not contributing to lysis, but because of the narrow focus of the study on the outer membrane it seems they are de facto interpreting lysis as exclusively due to mechanical destabilization of the outer membrane. In principle, this could be true, but it would be surprising (to me) if there were not other harmful

physiological changes occurring simultaneously with damage to the outer membrane (e.g. changes to peptidoglycan structure, free radicals, protein stress, membrane depolarization, PG and OM synthesis, etc.) that also contribute to cell death via a mechanism that may or may not involve the outer membrane. If they measured these and found one of them to also correlate with lysis progression, it is unclear to me whether or not they would still say that complement "kills bacteria by mechanical destabilization." This is important because the time scales of lysis are quite long (tens of minutes in many cases). At these time's hard to imagine that there is not a global dysbiosis occurring in the cell.

This is primarily as an interpretation/communication issue rather than a critique of the results. In my opinion, the manuscript does not quite show that "mechanical destabilization kills bacteria when they are subjected to complement" it shows that "the outer membrane becomes mechanically destabilized when bacteria are subjected to complement, which contributes to lysis." If the authors really wanted to show that outer membrane strength is the key limiting factor in complement-mediated killing, they would have to predictively tune outer membrane strength (through mutations to outer membrane composition) or the ability of complement to destabilize it (without, somehow, inhibiting its ability to form complexes), and then show that this meaningfully altered the ability of complement to kill cells.

We agree with the reviewer that it is essential that the interpretation and communication of our manuscript strictly align with our experimental data. It appears to us, however, that this reviewer comment originates in a misreading of our conclusions: we do not interpret "lysis as exclusively due to mechanical destabilization of the outer membrane" as here written by the reviewer, and do not claim to show "that outer membrane strength is the key limiting factor in complement-mediated killing".

We apologize if this was not sufficiently clear and would welcome suggestions to communicate this better, but first note that the title and abstract of the manuscript explicitly refer to "mechanical destabilization of the cell envelope" [not: "of the outer membrane"], where the (Gram-negative) cell envelope includes the outer membrane, the cell wall, and the inner membrane, as also noted in the first sentence of the introduction.

In our discussion, we explicitly mention other parts of the cell envelope, such as the (peptidoglycan) cell wall. As written in the discussion, "outer membrane poration will also cause leakage of

periplasmic contents: our data suggest that this affects maintenance of the cell wall and hence its mechanical integrity”, which is entirely in line with the reviewer’s comments above, and not attributing cell death to outer membrane destabilization alone.

-As it stands, Figures 1-3 are qualitative and may or may not represent the behavior of an average cell. For me, it was somewhat difficult to piece together what would be happening to such a cell. For example, Figures 2 and S5 are the key figures showing that blurring of the AFM precedes lysis, but they occur on vastly different time scales, which is interesting but also confusing. There are no n numbers, statistics, or quantifications provided for these measurements.

These figures are not only qualitative but also contain clear quantitative features: For some of these, it makes sense to explicitly quantify them, e.g., the roughness of the images over time with quantifications now included in Fig. 2 and Supplementary Fig. 5.

For others, such quantification does not seem of much additional value (e.g., a transition of many observed MAC pores before cell death to a complete failure to detect any pores, a failure that coincides with cell death, as shown in Fig. 2, with independent repeat data in Supplementary Fig. 5); or it is simply not so clear what would be a physical/biological meaningful parameter to quantify (e.g., the observation of growing/propagating defects in Fig. 3, with two independent measurement series included in the two Supplementary Videos). Instead, throughout the manuscript, we include original data sets that represent multiple repeat experiments each on a single cell. This may be considered descriptive but nonetheless demonstrates the consistency and repeatability of our observations.

We agree with the reviewer that the different timescales are both interesting and confusing. We have not conclusively identified the reason for the differences in onset of cell envelope destabilization, but it may be due to differences in MAC deposition, which is highly varied as we have reported and quantified in Supplementary Fig. 1. We do see a consistent timescale between destabilization and cell death though: it is seconds between the loss of AFM contrast and inner membrane permeabilization. However, these images are extremely difficult to generate, requiring consistent, high-quality AFM on a cell that happens to die during the imaging. This limits the throughput of our experiments and thereby our ability to define, e.g., the rate of defect formation, minimum number of MAC pores required for cell death, RMS roughness change upon cell death, etc. for “an average cell”.

That said, after presenting Figs. 1-3, our manuscript goes on to investigate if there are any underlying reasons for the observed correlation between cell envelope destabilization and cell death: we find that cells swell before inner membrane permeation, which we do quantify in Fig. 4.

-Similarly, it would be very helpful to have the distribution of lysis times as assayed via fluorescence microscopy and any such temporal information about the temporal dynamics of blurring of the AFM since this correlation is the central point of the paper.

To gain any mechanistic understanding from the lysis times as measured over a large population of cells, we'd need to acquire the according AFM data of these cells. However, AFM can only image one cell at the time; it would take unpractically long to, e.g., > 1 day of continuous operation (with no automatic procedure available) to measure just 100 cells in a single experiment. And that optimistic time estimate is under the unrealistic assumption that that cells and AFM tip would remain unchanged over the long duration of such an experiment. This prevents us from making meaningful correlations of cell envelope destabilization and cell lysis at population level.

To circumvent this problem, our manuscript instead focuses on single-cell experiments, with early stages of the experiments as intrinsic negative controls for the observation occurring after complement exposure and immediately prior to, or upon cell lysis. Most importantly, this has also enabled us to 1:1 correlate changes at the cell surface with cell lysis, without the convolution due to the high cell-to-cell variability in MAC formation.

Fig. R1. Percentage of dead cells, measured by SYTOX staining, as a function of time, as recorded for the experiments shown in Fig. 4 and Supplementary Figs. 10, 11.

To illustrate this point, Fig. R1 shows some of the lysis data that was acquired in the experiments underpinning Fig. 4 (with the time axis referenced to the beginning of the AFM experiment, as explained following the next comment). These show the % of dead cells as a function of time and suggest a lysis time distribution with a width >10 mins. It is not so clear to us which hypothesis such an observation could help to (in)validate.

-In line with the points above: the authors argue that because cells "could show no staining for inner membrane permeation" upon complement treatment, that outer membrane perforation is not sufficient to lyse cells. However, in figures 2A (and maybe 1B?), there appear to be lysed cells at 0 minutes after complement treatment. It is unclear what this means for the model although it is very interesting and, once again, begs for quantification of lysis times (at least) and ideally some explanation of what is presumably a broad distribution of lysis times.

The lysed cell in Fig. 2A is not "at 0 minutes after complement treatment", but (see caption Fig. 2A) "at the beginning of imaging". Typical times between complement treatment and beginning of imaging (due to unavoidable delays in setting up, locating AFM tip on cell surface, etc.) were between 10 and 15 minutes. We have now amended the Materials and Methods section to clarify this. "AFM imaging was initiated 10-15 minutes after C9 incubation, with the first AFM images recorded shortly after."

The variability in timescales occurs between complement treatment and inner membrane permeation (cell lysis). While this variability is not fully understood, this may be due to the high variability in MAC deposition. As quantified in Supplementary Fig. 1, there is a large spread of the number of MAC pores observed per area, when comparing different cells that have undergone the same treatment, even in the same sample.

Importantly, given the observed cell-to-cell variability in MAC formation and lysis times, temporal correlations between changes in cell surface topography/mechanics and cell death will be blurred in population studies. To robustly identify correlations between MAC formation and cell death, we therefore need to consider such correlations at single-cell level, which – in our view – is what makes our study unique.

Minor Comments

-In Fig. 1 can you show the corresponding image of an untreated control cell so that the reader can directly compare without having to look for it in another reference. Can you also add the n number. Quantifying the blurriness and showing the time courses for more than 1 cell would be helpful.

In the new Fig. 1B, we have now included comparable images of the surface of an untreated cell and of a cell treated with complement up to C5, in addition to the previously included images of cells treated with complement up to C9, showing MACs.

Comparable roughness quantifications and time courses have now been shown for $n = 3$, shown separately in Fig. 2 and Supplementary Fig. 5. Supplementary Fig. 5 has also been updated to be in the same organization as Figure 2 to make it easier to directly compare.

-In Fig. 2, can you show an image from the untreated control for reference. Can you also add the n number. Quantifying the blurriness and showing the time courses for more than 1 cell would be helpful.

These are single-cell experiments as a function of time, where we monitor and report on the changes at the cell envelope that accompany (and/or lead up to) cell death. As such, the appropriate negative control for these changes is the cell envelope of the same, complement-treated cell as measured at least several minutes before cell death, not a different untreated (set of n) cell(s).

That said, an additional untreated control has now been included in Fig. 1; RMS quantification has now also been added to Fig. 2; and Supplementary Fig. 5 now includes 2 additional datasets (instead of 1 repeat included previously). Importantly, these independent repeats all show the same indicators of cell envelope destabilization for cells measured before and after cell death.

-In Fig. 3B, is it possible to show the untreated (pre-treatment) control from the same area? Can you also add the n number and note how long after complement treatment 0 minutes corresponds to?

Similar to our response above, we here note that these are images monitoring changes of the same area at the surface of the same cell as a function of time, hence implicitly controlled by referencing to the first image (0 min) in the series. As Fig. 3B refers to a single-cell data set, it is not so clear which n number the reviewer refers to. An independent repeat experiment has been shown in Supplementary Fig. 8.

We apologize for lack of clarity on the time between complement treatment and the start of imaging. This has now been clarified in the Materials and Methods.

Can you also label the MACs and the defects on the image, for a while I assumed the defects were the MACs.

The MACs are labelled in Fig. 1. As a guidance to the reader, we have now also labelled larger defects in Fig. 2 as a guide to the reader. In addition, we have included further clarification in the text describing the MACs and larger defects.

-Fig. S5 seems to belong in the main text.

Supplementary Fig. S5 shows the same phenomena as the main Fig. 2 and was included in the SI to demonstrate repeatability of our observations, not to provide new information compared with Fig. 2. While we do not have a strong opinion on this, we therefore suggest leaving this as a supplementary figure, or refer to and next follow editor's guidance on this.

-In Fig. 4D, it is not useful to normalize to zero since now there is no information in the control bars (and no bars). If it's internally controlled in each experiment then it is more standard to normalize to 1. 5-10% difference for these variables is relatively large.

We think this comment is due to a misunderstanding of Fig. 4D. The reported changes in cell height and surface stiffness are a comparison between before and after cell death of the same cell, showing that the stiffness and height both increase as the cell dies, with the start of the increase typically detected before inner membrane permeation. The -C8,9 control (with a value close to zero) is the change in height and stiffness over the same time period in absence of cell death. So there is no normalization here, the negative control is just very consistently close to zero. We have tried to make this clearer in the figure legend and apologize for the misunderstanding.

Dear Bart,

Thank you for submitting a revised version of your manuscript. I sincerely apologise for the protracted assessment process due to delays in referee report submission. We have now received input from one of the original reviewers, who finds that their previous concerns have been addressed satisfactorily.

Therefore, there now remain only a few editorial points that need addressing before I can extend official acceptance of the manuscript:

1. Please check the spelling of the author's name Christian Bortolini in our submission website vs the manuscript.
2. Please check that the funding information is correct and identical both in the manuscript and our online system. Currently, the UK Department for Science, Innovation and Technology is missing in our system.
3. Please submit up to five keywords.
4. CRedit has replaced the traditional author contributions section because it offers a systematic, machine-readable author contributions format that allows for more effective research assessment. Please remove the Authors Contributions from the manuscript and use the free text boxes beneath each contributing author's name in our online submission system to add specific details on the author's contribution. More information is available in our guide to authors.
5. Please rename "Conflict of interest statements" section into "Disclosure and competing interests statement" (further info: <https://www.embopress.org/page/journal/14602075/authorguide#conflictsofinterest>).
6. Please rename "Material and Methods" section into "Methods".
7. Please rename the movies into Movie EV1-EV2 and update the callouts accordingly. The movie legends should be removed from the manuscript text file and provided in separate readme.txt files; then each movie should be zipped up with its legend and uploaded as folder per movie (Movie EV1, Movie EV2). Further information is available here: <https://www.embopress.org/page/journal/14602075/authorguide#expandedview>
8. In the Appendix, please add page numbers and a brief table of contents. Only 14 Appendix figures appear to be provided, while the front page of the Appendix mentions 15 figures. Please check.
9. Our data editors have flagged the following issues in figure legends that need correcting:
 - Please provide the exact p values in the legends of figures 4e-f.
 - Please define the measure of center for the error bar in the legend of figure 4d.
 - Please define that the white arrows in the legend of figure 2c.
10. Papers published in The EMBO Journal are accompanied online by a 'Synopsis' to enhance discoverability of the manuscript. It consists of A) a short (1-2 sentences) summary of the findings and their significance, B) 3-4 bullet points highlighting key results and C) a synopsis image that is 550x300-600 pixels large (width x height, jpeg or png format). You can either show a model or key data in the synopsis image. Please note that the image size is rather small and that text needs to be readable at the final size. Please send us this information together with the revised manuscript.

With best wishes,

leva

leva Gailite, PhD
Senior Scientific Editor
The EMBO Journal
Meyerhofstrasse 1
D-69117 Heidelberg
Tel: +4962218891309
i.gailite@embojournal.org

We realize that it is difficult to revise to a specific deadline. In the interest of protecting the conceptual advance provided by the work, we recommend a revision within 3 months (30th Nov 2024). Please discuss the revision progress ahead of this time with the editor if you require more time to complete the revisions. Use the link below to submit your revision:

Referee #3:

The manuscript is valuable and is suitable for publication. The images will be a valuable resource for medical microbiologists.

My main comments were addressed. One relatively minor point: the authors did not change much in response to the reviews. In their response they emphasize that the controls for the temporal dynamics of the effects of MACs are just the pre-treated cells even though the proper controls are the temporal dynamics of untreated cells. However, since the qualitative effects of the MACs are quite dramatic, the controls as they stand are sufficient even if the proper controls would be preferable. Finally, I would include the data in Fig. R1 somewhere in the paper and explicitly discuss the dynamics of lysis since this gives the reader real intuition for the process.

Editorial points that needed addressing:

1. Please check the spelling of the author's name Christian Bortolini in our submission website vs the manuscript.

Thanks for spotting the typo. It is corrected now.

2. Please check that the funding information is correct and identical both in the manuscript and our online system. Currently, the UK Department for Science, Innovation and Technology is missing in our system.

Thanks for spotting the omission. It is corrected now, with this funder now also listed on the online submission system.

3. Please submit up to five keywords.

Keywords have now been inserted in the manuscript, following the abstract, as follows:

Complement; Membrane Attack Complex; Escherichia coli; Atomic Force Microscopy; Bacterial Membranes.

4. CRediT has replaced the traditional author contributions section because it offers a systematic, machine-readable author contributions format that allows for more effective research assessment. Please remove the Authors Contributions from the manuscript and use the free text boxes beneath each contributing author's name in our online submission system to add specific details on the author's contribution. More information is available in our guide to authors.

The Author Contributions section has been removed from the manuscript; relevant information on author contributions is indicated with each author in the online submission system.

5. Please rename "Conflict of interest statements" section into "Disclosure and competing interests statement" (further info: <https://www.embopress.org/page/journal/14602075/authorguide#conflictsofinterest>).

Done.

6. Please rename "Material and Methods" section into "Methods".

Done.

7. Please rename the movies into Movie EV1-EV2 and update the callouts accordingly. The movie legends should be removed from the manuscript text file and provided in separate readme.txt files; then each movie should be zipped up with its legend and uploaded as folder per movie (Movie EV1, Movie EV2). Further information is available

here: <https://www.embopress.org/page/journal/14602075/authorguide#expandedview>

Done.

8. In the Appendix, please add page numbers and a brief table of contents. Only 14 Appendix figures appear to be provided, while the front page of the Appendix mentions 15 figures. Please check.

Page numbers were already in place. Table of contents has been provided at the beginning of the Appendix, now listing 14 instead of the incorrectly indicated 15 figures.

9. Our data editors have flagged the following issues in figure legends that need correcting:

- Please provide the exact p values in the legends of figures 4e-f.

These have now been added at the end of the figure caption.

- Please define the measure of center for the error bar in the legend of figure 4d.

Now clarified in the figure caption (D).

- Please define that the white arrows in the legend of figure 2c.

This has now been clarified in the figure caption, by referring back to figure 2A.

10. Papers published in The EMBO Journal are accompanied online by a 'Synopsis' to enhance discoverability of the manuscript. It consists of A) a short (1-2 sentences) summary of the findings and their significance, B) 3-4 bullet points highlighting key results and C) a synopsis image that is

550x300-600 pixels large (width x height, jpeg or png format). You can either show a model or key data in the synopsis image. Please note that the image size is rather small and that text needs to be readable at the final size. Please send us this information together with the revised manuscript.
This has now been provided.

Dear Bart,

Thank you for addressing the final editorial issues. I sincerely apologise for the delay in the processing of your manuscript due to the unusually high number of submissions that we receive at the moment. I am now pleased to inform you that your manuscript has been accepted for publication.

Before we forward your manuscript to the publishers, I would like to suggest minor edits in the manuscript abstract and synopsis. I have also written a blurb that will accompany the title of your manuscript on our website. Please take a look and let me know if any further edits are needed.

Blurb:

Atomic force microscopy analysis shows that the large-scale disruption of the E. coli outer membrane by membrane attack complex pores causes cell swelling and subsequent inner membrane damage.

Synopsis:

Complement proteins form membrane attack complex pores in the outer membrane of Gram-negative bacteria, leading to bacterial lysis via an incompletely understood mechanism. In this study, atomic force microscopy identifies a pathway of bacterial killing caused by membrane attack complex pore formation in the Gram-negative bacterial outer membrane.

- Atomic force microscopy shows formation of membrane attack complex pores on living E. coli cells.
- Initial pore formation leads to propagating defects and fractures in the outer membrane.
- Resulting cell death follows after overall mechanical destabilization of the cell envelope.

Abstract, 4th sentence:

"Initially, bacteria survived despite the formation of hundreds of MACs that were randomly distributed over the cell surface."

Finally, would it be possible to introduce some labelling in the synopsis image? While I very much like the minimalistic style, for our less expert readership it would be helpful to label, e.g., the MAC pores, inner and outer membrane. You can send me the edited file via email.

If you have any questions, please do not hesitate to contact the Editorial Office. Thank you for this contribution to The EMBO Journal and congratulations on a nice study!

Best wishes,

leva

leva Gailite, PhD
Senior Scientific Editor
The EMBO Journal
Meyerohofstrasse 1
D-69117 Heidelberg
Tel: +4962218891309
i.gailite@embojournal.org

>>> Please note that it is The EMBO Journal policy for the transcript of the editorial process (containing referee reports and your

response letter) to be published as an online supplement to each paper. If you do NOT want this, you will need to inform the Editorial Office via email immediately. More information is available here: https://www.embopress.org/transparent-process#Review_Process
